# Moderating Effects of Financial Cognitive Abilities and Considerations on the Attitude–Intentions Nexus of Stock Market Participation

Tahmina Akhter [1,*] and Mohammad Enamul Hoque [2]

1   Department of Finance, University of Dhaka, Dhaka 1000, Bangladesh
2   BRAC Business School, BRAC University, Dhaka 1212, Bangladesh; iiuc.enam@ymail.com
*   Correspondence: tahmina_akhter@du.ac.bd; Tel.: +880-1746086320

**Abstract:** This study aims to examine the determinants of investors' behavioral intentions to participate in the stock market. In this attempt, this research investigated the direct and moderating effects of the financial cognitive abilities and the financial considerations on the nexus of attitudes and behavioral intentions of investors. Data for this study were collected from active and potential investors in the Dhaka Stock Exchange of Bangladesh using a structured questionnaire. The partial least squares method was used to examine the nature and extent of the relationships of investors' behavioral intentions with their attitude, financial cognitive abilities, and financial considerations in making stock market investment-related decisions. The findings of this study suggest that investors' attitudes, financial planning ability, and perceptions of financial risks and benefits are important factors that influence their decisions in stock market participation. Moreover, financial planning, financial satisfaction, and perceived financial risk moderate the nexus of attitude and behavioral intentions to participate in the stock market. This study, therefore, has significant implications for policymakers, stock market regulators, and financial service providers.

**Keywords:** investors' attitude; behavioral intentions; financial risk perception; financial benefit perception; financial satisfaction; financial planning; stock market participation

## 1. Introduction

Capital market theory suggests that every household should invest a part of their wealth in risky assets, for example, in equities, to earn a return with the risk premium on their investments (Campbell et al. 2003; Curcuru et al. 2010; Markowitz 1952). However, the empirical evidence on households' portfolio choice patterns indicates that many of them do not invest in the share market, which is referred to as the "stock market participation puzzle" (Christelis et al. 2010; Guiso et al. 2003). According to a number of studies, the fixed monetary participation cost (Calvet et al. 2007; Vissing-Jørgensen and Attanasio 2003) in the stock market plays a pivotal role in non-participation, whereas Andersen and Nielsen (2010) concluded that the behavioral biases and cognitive skills of individuals seem to influence their decision to participate in the stock market. Behavioral finance also supports that individuals' thinking processes and cognitive errors (Mate and Dam 2018) affect their financial investment-related decision-making process. Often, stock market participants follow "bounded rationality", which refers to decisions that are satisfactory to themselves rather than taking optimal decisions as suggested by the "rational expectations" theory (Simon 1955). Additionally, in the investment decision-making process, investors' beliefs or cognitive abilities (Dimmock and Kouwenberg 2010), personal traits (Van Rooij and Teppa 2014; Wäneryd 2001), behavioral preferences (Georgarakos and Pasini 2011), and their past experience (Lo 2005) have been found to play a significant role towards the intention of stock market participation. Participation of individuals in the stock market is a crucial part of economic growth for any developing country (Bekaert and Harvey 1998; Özbilgin 2010).

Therefore, to boost the performance of the stock market of a country, market regulators and the government must understand how the attitude and intention of the market participants influence their investment decisions in the capital market.

Over the past few decades, various empirical studies attempted to investigate the psychology behind investing in the stock market. Earlier research in the behavioral domain hypothesized that attitudes were the primary predictors of human behaviors and assumed that attitudes have a direct impact on an individual's behavioral intentions to perform a task or make a decision (Vroom 1964; Wicker and Pomazal 1971). The empirical evidence suggests that the level of financial literacy, which refers to the ability to understand and use various financial skills, of investors has both a direct influence and an incremental impact on their attitude and intentions to participate in the stock market (Baker and Ricciardi 2014; Cole et al. 2012; Von Gaudecker 2015; Hadi 2017; Howlett et al. 2008). Another important part of cognitive skills is to formulate a plan about managing financial assets that will contribute to the prosperity of individuals (Perry and Morris 2005). Financial plans guide future financial decision making in rapidly changing external circumstances and, therefore, it influences the behavioral intentions of individuals to make certain financial decisions, for example, in order to formulate a portfolio, one must know how much to invest in equities and how much to invest in bonds (Yeske and Buie 2014). Additionally, according to prior studies, the level of financial satisfaction, which refers to people's subjective evaluation of their financial situations, influences the financial behavior of individuals (Atlas et al. 2019; Durand 2015; Rao et al. 2016). Similarly, risk attitudes, for example, "negative wealth shock", "uncertainty dispersion", "investors", "affinity to bet", "religion-incited betting attitude", "hedging potential", and "corporate extortion disclosures", can impact the decisions of market participants (Barsky et al. 1997; Dimmock and Kouwenberg 2010; Giannetti and Wang 2016; Kumar et al. 2011). Studies on investors' risk perceptions linked to their financial behavior have confirmed that individuals' risk-taking attitudes in other aspects of their lives can influence their risk-related behavior in financial decisions (Barsky et al. 1997).

The growing empirical evidence of inconsistency between attitudes and behaviors (Wicker 1969) has encouraged researchers to investigate the moderating effects of various situational, conditional, contextual, and personal factors on the relationship between attitude and behavior (Ajzen and Fishbein 2005). Studies have indicated that when cognitive skills or abilities, for example, obtaining information from one's surroundings, understanding the situation on the basis of information, planning how to act, and executing a behavior (Shinohara 2016), interact with attitudes, they can better explain the behavioral intentions in financial decision making (Agarwal and Mazumder 2013; Ali et al. 2021; Hayat and Anwar 2016; Kaur and Arora 2021; Nadeem et al. 2020). Moreover, since investment decisions in the stock market are financial decision that concern asset accumulation for welfare and consumption smoothing (Cole et al. 2012); consequently, the successful planning for future financial investment and the perception of well-being achieved from those decisions should affect the behavioral intentions of the individuals in stock market participation. Few studies to date, however, have sought to investigate the impact of financial planning and financial satisfaction on the nexus of investors' attitudes and behavioral intentions for stock market participation. Another key determinant of investment decision making in the stock market is the perception of individuals regarding risk. Nevertheless, like any other financial decision, stock market investment decisions involve both risks and benefits. According to a study by Liu et al. (2013), perceptions regarding price benefits and convenience can influence an individual's financial decisions; therefore, it can be inferred that investors' beliefs regarding potential benefits are likely to influence their investment decisions in the stock market. Although there are several studies (Nadeem et al. 2020; Shehata et al. 2021) that have investigated the interaction effect of the risk perceptions on the attitudes and behavioral intentions of stock market participants, according to our best knowledge, very few studies have examined the moderating role of perceived benefits on the nexus of investors' attitude and behavioral intentions to participate in the stock market.

In line with the above discussion, this research investigates the impact of investors' attitudes on their behavioral intentions along with how the above conceptualized variables moderate the relationship nexus between "attitude and behavioral intentions" of stock market participation. In doing so, the current study makes several contributions to the existing body of literature on investors' "attitudes and behavioral intentions" towards stock market participation. First, unlike any other previous study, this research offers a novel approach by integrating the theories of planned behavior, the theory of well-being, and the theory of risk perception attitude in the context of investors' behavioral intentions regarding stock market participation. The estimated model of this study captured 68.88 percent of the variations in behavioral intentions of stock market participation, which postulates an excellent degree of statistical acceptability. Hence, the integrated conceptual model will facilitate future studies to consider a diverse population of investors from other stock markets. Second, though literature suggests that "financial satisfaction" and "financial planning" are important determinants of financial decision making (Ali et al. 2015; Atlas et al. 2019; Koropp et al. 2014; Xiao and Porto 2017); empirically, it is less understood how these two variables moderate the relationship between investors' "attitude and behavioral intention" towards stock market participation. In this case, our findings add valuable insights into the existing body of literature regarding the interaction effects of financial planning and financial satisfaction on investors' attitude and behavioral intention nexus of stock market participation. Third, academics and researchers have widely investigated the relationship between investors' risk attitudes and financial decision making (Fellner-Röhling and Maciejovsky 2007; Oehler et al. 2018; Weber et al. 2002; Weber 2010) and how the risk perceptions of individuals moderate the relationship between their attitude and intentions to participate in the market (Nadeem et al. 2020; Shehata et al. 2021). However, few studies have examined how investors' perceptions regarding the benefits moderate their attitude pertaining to stock market participation. Therefore, the findings of this study will highlight the fact of whether investors' perceptions regarding the potential benefit from a risky investment is of similar importance such as the perceived risk when they shape their behavioral intentions to invest in the stock market.

The study results show that investors' attitudes, ability to financially plan, and perceptions regarding risks and benefits are important predictors of their behavioral intentions for stock market participation. As conceptualized by our research model, the investors' financial planning abilities, their financial satisfaction levels, and risk perceptions were found to have moderating effects on the relationship nexus of attitude and behavioral intentions when they considered making investment-related decisions in the stock market. Although the control variable educational level used in this study was found to have a significantly positive influence on the investors' behavioral intentions to participate in the stock market, according to the findings of this study, the financial literacy level neither had any direct influence on the behavioral intentions nor had any incremental impact on the nexus of investors' attitudes and behavioral intentions towards stock market participation. A plausible explanation behind this could be that investors perceive the stock market to be too volatile and, in spite of possessing the necessary financial knowledge, they do not have the confidence to take the investment decisions in the stock market (Waheed et al. 2020).

The rest of the study is presented as follows: Section 2, the "Literature Review", presents the supporting literature of the variables utilized in this study, the hypotheses of the study with theoretical and empirical arguments along with the research framework. Section 3, the "Methodology", enumerates the sampling, data collection, research design, and model estimation with empirical results. Section 4, the "Empirical Results Discussion", presents the main findings of the study along with their implications. In Section 5, the "Summary and Implications", we conclude the study with practical implications, limitations, and scopes for future research.

## 2. Literature Review

### 2.1. Attitude and Behavioral Intentions towards Stock Market Participation

In 1975, Ajzen and Fishbein proposed the "theory of reasoned actions" (ToRA), which attempted to establish the relationship between individuals' attitude and behavioral intentions. Based on ToRA, Ajzen (1991) developed the "theory of planned behavior" (TPB) that determines the intentions of an individual to engage in a specific behavior, which can be explained by his/her motivations and ability or behavioral control. Since the TPB designates that an individuals' attitude, subjective norms, and perceived behavioral control notably affect their behaviors and behavioral intentions, the current study utilized this theory to predict investors' attitudes towards participation in the stock market. In financial decision making, the attitudes of individuals are related to their behavioral intentions (Hoque et al. 2019; Koropp et al. 2014). According to the TPB (Ajzen 1991), attitudes are defined as "the degree to which an individual derives a positive or negative valuation from performing a specific behavior". Therefore, the "attitude" can explain investors' favorable or unfavorable assessments regarding their intentions to invest in the market.

The empirical evidence demonstrates that attitude towards investment decisions is a crucial factor that influences stock market participation decisions (Klontz et al. 2011; Shih and Ke 2014). Moreover, prior studies have confirmed that investors' attitudes towards investment, "subjective norms", and "perceived behavioral control" positively influences their behavioral intentions to participate in the stock market (Nadeem et al. 2020; Phan and Zhou 2014). Based on these findings on stock market participation, the current study hypothesized the following:

**Hypothesis 1 (H1).** *Investors' attitudes have positive influence on their behavioral intentions to participate in the stock market.*

### 2.2. The Moderating Effects of Cognitive Abilities

#### 2.2.1. Financial Literacy

According to standard portfolio choice models, it is expected that individuals with financial knowledge can make rational investment decisions to maximize their utilities over their lifetime. Since financial literacy is acquired by people with certain personality traits, for example, people with high "intellect" and determination to control their economic well-being, they are therefore more likely to invest in improving their knowledge (Lusardi and Mitchell 2008; Pinjisakikool 2017). These people tend to make informed investment decisions, as they are able to easily collect information at low cost, and they tend to seek professional financial advice. On the other hand, a lack of financial knowledge can lead to delegation of the decision making or complete avoidance of risky investment decisions (Calcagno and Monticone 2015; Von Gaudecker 2015). Moreover, it can be inferred from "perceived behavioral control" of the TPB that the level of ease or difficulty in financial decision-making affects investors' intentions towards stock market participation. Therefore, it can be presumed in line with the theory that individuals' level of financial education will impact their behavior and intentions to invest in the market. In this context, a number of studies examined the direct influence of financial literacy on the investment decisions of the individuals and concluded that financial knowledge significantly affects risky financial decisions (Kumari 2020; Mandell and Hanson 2009; Al-Tamimi and Kalli 2009; Van Rooij et al. 2007).

However, interest on the moderating role of financial literacy/knowledge is growing (Hayat and Anwar 2016; Hadi 2017; Nadeem et al. 2020), and the interaction of financial literacy and attitude could be interesting to investigate. An individual with some financial literacy/knowledge struggles less and also pays less for gathering and handling information compared to an individual with low financial literacy/knowledge. Henceforth, when both knowledge and attitudes interact, that may change behavior, and as a result financial literacy could moderate the existing relationship between attitude and behavioral intention in stock market participation. Several prior studies have also discovered that

financial knowledge plays a moderating role in the relationships between behavioral biases and investment decisions (Hayat and Anwar 2016), emotional intelligence and investment decisions (Hadi 2017), demographic characteristics and financial risk tolerance (Shusha 2017), and money attitude and stock market participation (Nadeem et al. 2020). Therefore, based on the empirical evidence, this study hypothesized that:

**Hypothesis 2 (H2).** *Financial literacy positively influences individuals' behavioral intention to participate in the stock market.*

**Hypothesis 2a (H2a).** *Financial literacy intensifies the relationship between individuals' attitude and their behavioral intention to participate in the stock market.*

### 2.2.2. Financial Planning

Financial planning by individuals incorporates their current financial condition and long-term monetary goals; whereas investment planning can be considered as a major component of financial planning to achieve short- and long-term goals. Financial planning involves both "cognitive factors" (mental process) and "affective" (emotional) issues (Baker and Ricciardi 2014). Therefore, when individuals translate their investment goals into investment decisions, their attitude and belief system are expected to influence their investment decisions. According to the TPB, "perceived behavioral control" can directly influence the behavior of individuals. When they have easy access to the control factor, they feel motivated to utilize the resource to obtain the expected outcome. Individuals perceive that the profitable investment decision is an outcome of proper financial planning (Asandimitra et al. 2019). Additionally, a study on professionals by Arpana and Swapna (2020) indicated that the relationship between financial planning and financial behavior was significant. Hence, when investors are capable of financial planning, it can positively influence their behavioral intention towards investing in the stock market. Based on the evidence on financial planning, there is a need to examine the moderating role of financial planning on the relationship between investors' attitude and behavioral intention of stock market participation. Accordingly, this research hypothesized:

**Hypothesis 3 (H3).** *Financial planning positively impacts investors' behavioral intention of stock market participation.*

**Hypothesis 3a (H3a).** *Financial planning intensifies the relationship between investors' attitude and behavioral intentions to participate in the stock market.*

### 2.2.3. Financial Satisfaction

In the capital market, individuals make investment decisions to achieve their desired level of financial satisfaction. According to the utility maximization principle of economics, individuals make rational choices while taking any decision. However, the subjective "theory of well-being" proposed by Wilson (1967) states that, "how well our lives go for us is a matter of our attitudes towards what we get in life rather than the nature of the things themselves". Past research utilized objective proxies of well-being, namely, income, literacy, life expectancy, financial satisfaction, etc., and found that the level of well-being varied among individuals depending on their subjective evaluation, for example, perceptions (Durand 2015). As a result, it can be assumed that when individuals decide to participate in the capital market, their perceived level of well-being and satisfaction influence the choices of investment decisions they make. In the literature, financial satisfaction is found to have direct and significant impacts on financial knowledge, risk tolerance, and investor behaviors (Atlas et al. 2019; Joo and Grable 2004; Parmitasari et al. 2018; Sahi 2017). However, to date, few studies have investigated the moderating role of the financial satisfaction of investors on the nexus of attitude and behavioral intentions of stock market participation. Therefore, based on the above discussion the current study hypothesized the following:

**Hypothesis 4 (H4).** *Financial satisfaction positively affects participants' behavioral intention to participate in the stock market.*

**Hypothesis 4a (H4a).** *Financial satisfaction intensifies the nexus of attitude and behavioral intentions of stock market participation.*

*2.3. The Moderating Effects of Financial Considerations*

2.3.1. Perceived Financial Risk

According to the risk–return framework, individuals' perception of risk attitude refers to the positive or negative weight assignment towards a risky financial decision to determine the level of the option's desirability. In the ToRA framework (Ajzen and Fishbein 1975), therefore, preference for a risky investment represents the attitude of the individuals towards the risky option. However, the conceptualization of "attitude" and its impact on behavioral intentions of investment decision making is somewhat different in this research. The perception towards the risk attitude of this study can be best explained by the "risk perception attitude" (RPA) theory developed by Rimal and Real (2003). RPA posits that the risk perceptions of individuals motivate their behavioral action, and the efficacy beliefs are critical for facilitating changes in behavior. Therefore, under the framework of RPA, the behavioral intentions of the stock market participants can be expected to be influenced by their risk perceptions regarding their investment decisions in the stock market.

The literature on stock market participation confirms the existence of the significant impact of risk attitude on the investment decisions of individuals (Barasinska et al. 2012; Fellner-Röhling and Maciejovsky 2007; Noussair et al. 2014; Weber 2010; Zhang et al. 2019; Zhang and Li 2011). Moreover, the attitude of investors towards risky investments and stock market participation were affected by their personality type (Rao et al. 2016). Additionally, individuals' ability to cover the associated financial risk has a significant impact on their investment decision (Cocco et al. 2005; Heaton and Lucas 2000; Niu et al. 2020). Studies indicate that the degree of risk taking (Clark-Murphy and Soutar 2004), risk seeking (Keller and Siegrist 2006), and risk tolerance (Cole et al. 2012) have significant influence on investors' decisions to participate in the stock market. Several studies have investigated the moderating role of risk perception on the relationship between individuals' attitude and behavioral intentions towards financial decisions and found that the risk perception negatively influences this relationship nexus (Hoque et al. 2019; Kaur and Arora 2021). Therefore, it can be deduced that if the individuals perceive that high risk is related to their investment decision in the stock market, it can negatively influence their intention to participate in the market. In line with the above discussion the current research hypothesized the following:

**Hypothesis 5 (H5).** *Perceived risk negatively affects participants' behavioral intention to participate in the stock market.*

**Hypothesis 5a (H5a).** *Perceived risk weakens the nexus between attitude and behavioral intentions of stock market participation.*

2.3.2. Perceived Financial Benefit

Perceived benefits are the beliefs of the individuals concerning positive outcomes of a specific behavioral action (Chandon et al. 2000). The RPA theory suggests that individuals' perceptions of benefits concerning their decisions play a significant role in the likelihood of their behavior towards an action (Rimal and Real 2003). In investment decision making, the "perceived benefit", therefore, postulates the satisfaction and the positive outcomes that an individual expects to yield by engaging in an investment decision in the stock market. Investors' past experience regarding stable dividend income and capital gain from stock market investment tend to influence their perception regarding future returns (Ganzach 2000) from their intended investment decision. The literature suggests that

when individuals perceive to attain some benefit from a financial decision, their perception positively influences their attitude towards that decision (Ali et al. 2021; Liu et al. 2013). Moreover, if the individuals believe that the benefits of the respective financial decision outweigh the costs, that perception positively moderates the relationship between their attitude and behavioral intention of the financial decision (Hoque et al. 2019). Therefore, it can be similarly inferred that the perception of the potential benefits of investment decisions in the stock market will favorably influence the existing relationship between customers' attitude and their behavioral intention to participate in the market. Based on the above discussion, the current study hypothesized that:

**Hypothesis 6 (H6).** *Perceived benefit positively influences participants' behavioral intention to participate in the stock market.*

**Hypothesis 6a (H6a).** *Perceived benefit intensifies the relationship between attitude and behavioral intentions to participate in the stock market.*

### 3. Methodology
#### 3.1. Study Sample: The Case of Dhaka Stock Exchange, Bangladesh

The current study investigated the moderating roles of financial cognitive abilities and financial considerations on the relationship between attitude and behavioral intentions of stock market participation. To achieve the research objectives, this study utilized the Dhaka Stock Exchange (DSE) as a sample to collect data from actual and potential investors of the stock market. DSE, the major stock exchange in Bangladesh, is one of the smallest markets in Asia, but it is the third largest bourse in South Asia. DSE uses a highly fault tolerant automated system to facilitate the efficient and smooth trading of shares, debentures, and a variety of other securities for market participants. Nevertheless, DSE is considered a developing market; hence, this present study offers some added significance to the existing body of knowledge. For example, the DSE's characteristics are unique in terms of its legal framework, regulatory framework, quality of governance, market norms, and investor preferences (for detailed explanations, please see Boubaker et al. (2012) and Boubaker and Nguyen (2014a, 2014b)). Such markets require special attention, since there is a chance of obtaining empirical evidence that goes beyond the outcomes of the usually studied equity markets (Bekaert and Harvey 2002). Compared to investors in developed markets, those in developing markets are more likely to have behavioral biases in their investment decisions (Kristoufek and Vosvrda 2013; La Porta et al. 2000; Ng et al. 2016). Moreover, prudent financial decision making is an outcome of improved economic behavior that is evident in market participants who possess financial literacy and basic financial knowledge (Howlett et al. 2008). Since in the DSE, the current financial literacy level of individual investors is only 28 percent (source: Digital Financial Service Lab of government of Bangladesh), the low level of financial awareness may result in irrational investment preferences (Mate and Dam 2018) and noise-based trading (Chen et al. 2007). Therefore, the empirical evidence of the current study, in the context of investors in a developing market, offers a diverse set of implications for policymakers and regulators that can help to boost participation in the stock market.

#### 3.2. Questionnaire and Measurement

To measure the behavioral intentions of stock market participation along with its predictors, we developed a structured questionnaire that consisted of two parts: measurement questions and a demographic information section. The questionnaire was composed of seven constructs that incorporated forty-four items including demographic factors of gender, age, education, and trading experience. The measurement scales used in this study were adopted from the existing literature and modified in the context of stock market participants in Bangladesh. To measure the attitude of the investors for making an investment decision in the stock market, the current study adopted five items from

Hoque et al. (2019) and one item from Ali (2010). The ten questions to measure basic financial literacy were adopted from Ali et al. (2015) and Lusardi and Mitchell (2007). The seven items of financial planning and six items of financial satisfaction were all adopted from Ali et al. (2015). The construct, perceived risk, had seven items, and all were adopted from Ali (2010). The four items of perceived benefit were adopted from Liu et al. (2013), and two items from Hoque et al. (2019). Finally, the five items for the construct, behavioral intentions, were adopted from technology acceptance studies (Hausman and Siekpe 2009; Kaplan et al. 2007). The variable, financial literacy, was measured against the scores received by the participants from a simple quiz containing ten basic financial literacy-related questions (Appendix A). The literacy measures used in this study can be regarded as tier one or a basic financial literacy level as determined by Lusardi and Mitchell (2007). Since our target respondents were both actual and potential investors of DSE with diverse educational backgrounds, the quiz, though it may not be exhaustive, was sufficient to measure the literacy level of the respondents. All variables were measured using a five-point Likert scale, where one (1) represents "strongly disagree" and five (5) means "strongly agree". To check the depth, unambiguity, and simplicity of the respondents' answers to the questions, the questionnaire used in this study was tested for its reliability and face validity by academicians and practitioners. Based on their suggestions, the questionnaire was adjusted to make it easier to understand and smoother to read.

*3.3. Data Collection*

The sample for this study was composed of investors in the Dhaka Stock Exchange (DSE), Bangladesh, and the sampling was adopted from Kline (2011). Primary data for the study were collected from the active and potential investors in the DSE. Data were collected directly from investors who visited the brokerage houses. Due to the online trading facility and the COVID-19 pandemic, the number of investors who visited the stock exchange was quite low. As a result, the study utilized the convenience sampling technique based on the availability of the participants. A total of 386 valid questionnaires were received from respondents. According to Hair et al. (2013), this sample size can be considered sufficient, as their study suggested that 25 respondents per construct is necessary. Table 1 presents the demographic profiles of the respondents who participated in this study. Out of a total 386 respondents, 67.1 percent were male, and 32.9 percent were female. The male-to-female ratio of the participants was 2.04:1, that is, the number of males was almost double compared to female participants in this study. Most of the investors in this study belonged to the age group of twenty-six (26) to thirty-five (35) years and thirty-six (36) to forty-five (45) years. Together, these two age groups represented 67.1 percent of the participants. Moreover, most of the respondents, 77.2 percent, had attended some sort of higher education (graduation and post-graduation) institution.

**Table 1.** Demographic Profile.

| Category | Group | Numbers of Respondents | Frequency |
|---|---|---|---|
| Gender | Male | 259 | 67.1% |
| | Female | 127 | 32.9% |
| Age | 18–25 Years | 27 | 7.0% |
| | 26–35 Years | 139 | 36.0% |
| | 36–45 Years | 120 | 31.1% |
| | 46–55 Years | 58 | 15.0% |
| | 55+ Years | 42 | 10.9% |
| Educational Background | SSC | 23 | 6.0% |
| | HSC | 65 | 16.8% |
| | Honors | 184 | 47.7% |
| | Masters | 114 | 29.5% |

*3.4. Model and Empirical Estimation*

This study aimed to examine the influence of investor attitudes on the behavioral intentions in stock market participation and how the financial cognitive abilities and financial considerations of the participants affected the abovementioned relationship nexus. The conceptual research model used in this study is presented in Figure 1. The research hypotheses are indicated by the arrowed paths (the dotted arrows indicate a direct influence, and the solid arrows represent interaction effects) connecting the latent variables. The measures for the constructs used in this model were mainly adopted from the literature. Moreover, the measurement items are reflective in nature. The partial least squares (PLS) method was used to analyze the data as suggested by Hair et al. (2013), and Smart PLS 3.0 was utilized to investigate the nature and extent of the relationship among the variables in this study.

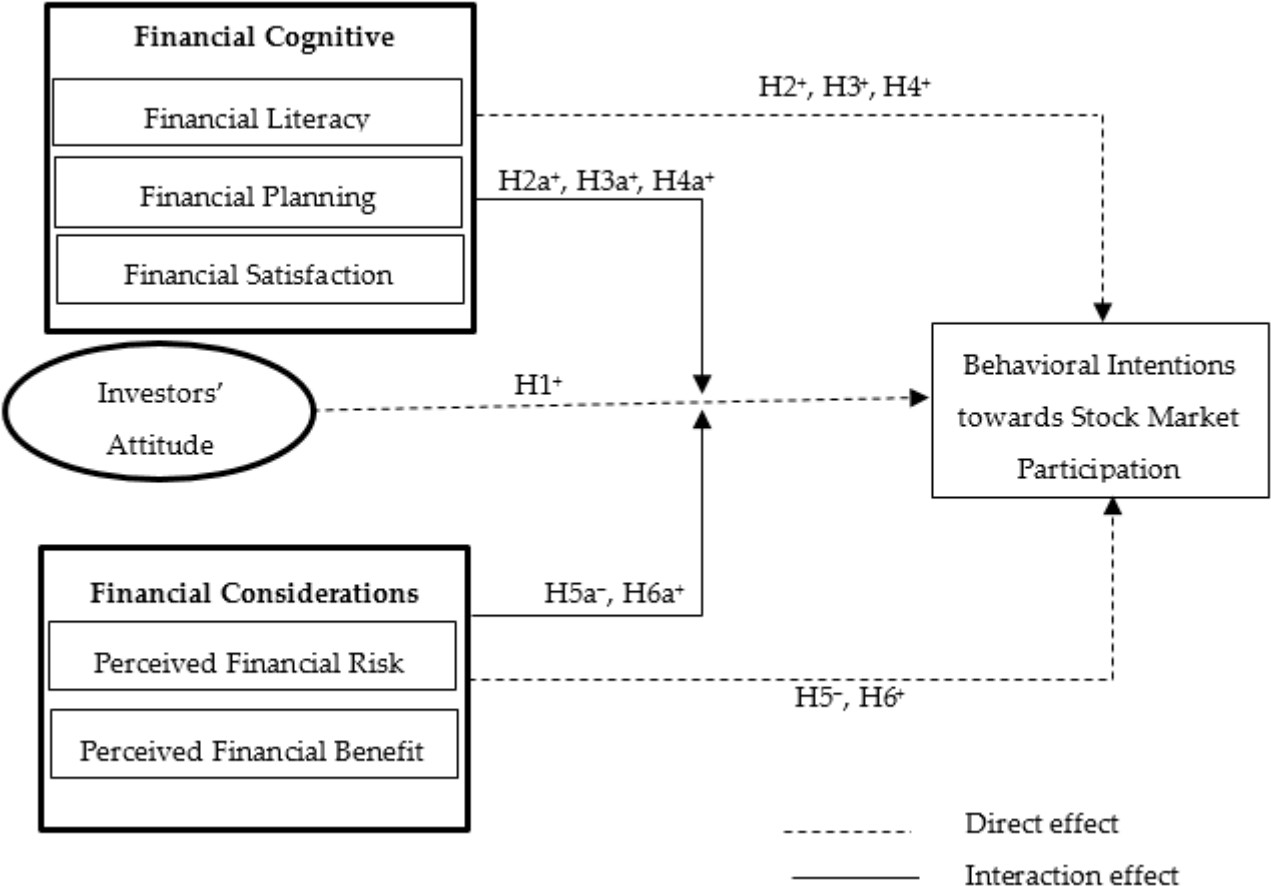

**Figure 1.** Conceptual research model.

3.4.1. Measurement Model Evaluation

Initially, the measurement model was evaluated to examine the discriminant and convergent validity of the observed variables. When the measurement items of a construct had a composite reliability and a Cronbach's alpha value of 0.70 or above, they indicated that a convergent reliability was established, and the construct had internal consistency. As shown in Table 2, all constructs' composite reliability values and the values of Cronbach's alpha were above the required threshold. Indicators with loads less than 0.5 were also eliminated, but some with loads between 0.50 and 0.70 were maintained since their removal had no effect on the average variance extracted (AVE) or composite reliability. Henceforth, measurement item loading shows that all measurements consistently represented the same latent construct (Hair et al. 2013). Accordingly, all constructs showed internal consistency and convergent reliability.

**Table 2.** Construct measurement items, reliability, and validity.

| Construct | Measurement Items | Factor Loading | Cronbach's Alpha (>0.70) | rho_A | Composite Reliability (>0.70) | AVE (>0.50) |
|---|---|---|---|---|---|---|
| Attitude (ATT) | ATT1: Choosing the stock market is a good idea | 0.707 | 0.839 | 0.866 | 0.878 | 0.509 |
| | ATT2: I like to invest in the capital market | 0.698 | | | | |
| | ATT3: Most people who are important to me have investments in the stock market | 0.674 | | | | |
| | ATT4: My family members prefer the stock market | 0.597 | | | | |
| | ATT5: I prefer the stock market because it is interest-free income | 0.794 | | | | |
| | ATT6: I prefer the stock market because of the capital gain opportunities | 0.825 | | | | |
| | ATT7: I can trade or invest in the stock market whenever I want | 0.674 | | | | |
| Perceived Benefit (PB) | PB1: The profit from the share market is higher than other investments | 0.719 | 0.852 | 0.862 | 0.894 | 0.628 |
| | PB2: Investing in the stock market seems to generate high returns for me (e.g., dividends and capital gains) | 0.798 | | | | |
| | PB3: I believe the stocks I invested in will perform satisfactorily in the future | 0.851 | | | | |
| | PB4: I think investing in stocks is highly rewarding | 0.811 | | | | |
| | PB5: Higher returns motivate me to invest in the share market | 0.777 | | | | |
| Perceived Risk (PR) | PR1: It is a risky decision to invest in the share market | 0.67 | 0.717 | 1.868 | 0.785 | 0.564 |
| | PR2: I may lose money due to the uncertainty in the stock market | 0.90 | | | | |
| | PR3: It is important to avoid monetary losses | 0.66 | | | | |
| Financial Literacy (FL) * | FL Is a single item construct with a score ranging from 0 to 10 | 1.000 | 1.000 | 1.000 | 1.000 | 1.000 |
| Financial Planning (FP) | FP1: I save money for retirement | 0.637 | 0.709 | 0.749 | 0.815 | 0.525 |
| | FP2: At any time, I have some money saved for emergencies | 0.717 | | | | |
| | FP3: I ensure that with every pay, I save some | 0.813 | | | | |
| | FP5: My insurance/takaful coverage is sufficient to meet costs related to emergency events | 0.722 | | | | |
| Financial Satisfaction (FS) | FS2: I am satisfied with my current financial situation | 0.756 | 0.835 | 0.838 | 0.883 | 0.602 |
| | FS3: I can do little to improve my current financial situation | 0.824 | | | | |
| | FS4: I rarely run short of money | 0.794 | | | | |
| | FS5: Based on my current financial situation, I could easily obtain a loan if I needed one (e.g., car loans, personal loans) | 0.786 | | | | |
| | FS6: If I had a major loss of income I could manage for a period of time (e.g., for 3 months) | 0.715 | | | | |
| Behavioral Intention (BI) | BI1: I will invest in the share market | 0.767 | 0.840 | 0.841 | 0.893 | 0.677 |
| | BI3: I will speak favorably about investing in the share market | 0.823 | | | | |
| | BI4: I will recommend investing in the share market if someone asks for my advice | 0.877 | | | | |
| | BI5: I will encourage my friends and family to invest in the share market | 0.819 | | | | |

A composite reliability of 0.70 is recommended (Hair et al. 2013), and Fornell and Larcker (1981) recommended a CR value of 0.60 or more and an AVE greater than 0.5. * FL has a single score based on the correct answers of quiz.

Additionally, the current study used the Heterotrait–Monotrait ratio (HTMT) and the Fornell–Larcker criterion to measure the discriminant validity of the measurement model of this study. The HTMT measures the similarity of the latent variables. If after running the bootstrapping routine of the HTMT inference, the results show a value below 0.90, the discriminant validity is satisfied (Henseler et al. 2015). Therefore, the HTMT ratio values presented in Table 3 among the pairs of latent variables indicate the discriminant validity for all pairs of variables in this study. The Fornell–Larcker criterion was utilized to assess

the degree of shared variance between each pair of the latent variables of the structural model of this study. Table 4 shows a comparison between the square root of the AVE of a construct and its bivariate correlation with other constructs; the higher AVE square root values of the variables, compared to the variables' correlation with other latent variables, confirms the discriminant validity of the measurement model of this research.

**Table 3.** Heterotrait–Monotrait Ratio (HTMT).

|      | ATT   | BI    | FL    | FP    | FS    | PB    |
|------|-------|-------|-------|-------|-------|-------|
| BI   | 0.768 |       |       |       |       |       |
| FL   | 0.042 | 0.045 |       |       |       |       |
| FP   | 0.509 | 0.494 | 0.233 |       |       |       |
| FS   | 0.351 | 0.255 | 0.096 | 0.405 |       |       |
| PB   | 0.744 | 0.676 | 0.037 | 0.487 | 0.276 |       |
| PR   | 0.164 | 0.126 | 0.045 | 0.120 | 0.182 | 0.097 |

ATT = attitude, PB = perceived benefit, PR = perceived risk, FL = financial literacy, FP = financial planning, FS = financial satisfaction, and BI = behavioral intention.

**Table 4.** Fornell–Larcker Criterion.

|      | ATT    | BI     | FL     | FP     | FS    | PB    | PR    |
|------|--------|--------|--------|--------|-------|-------|-------|
| ATT  | 0.714  |        |        |        |       |       |       |
| BI   | 0.667  | 0.823  |        |        |       |       |       |
| FL   | 0.003  | 0.005  | 1.000  |        |       |       |       |
| FP   | 0.402  | 0.406  | 0.191  | 0.725  |       |       |       |
| FS   | 0.291  | 0.216  | −0.032 | 0.331  | 0.776 |       |       |
| PB   | 0.640  | 0.580  | 0.013  | 0.384  | 0.230 | 0.793 |       |
| PR   | −0.082 | −0.121 | 0.000  | −0.016 | 0.012 | 0.048 | 0.751 |

ATT = attitude, PB = perceived benefit, PR = perceived risk, FL = financial literacy, FP = financial planning, FS = financial satisfaction, and BI = behavioral intention.

### 3.4.2. Structural Model Evaluation and Hypothesis Testing

Once the measurement model had met the required threshold, the structural model was tested. The $R^2$ (i.e., reliability indicator) for endogenous components was used to evaluate the structural model. The purpose of a variance analysis ($R^2$) or determination test is to identify the influence of exogenous factors on endogenous variables. In Figure 2, a 68.88 percent $R^2$ suggests that 68.88 percent of the variation of behavioral intention is explained by constructs and factors considered in the model, and the remaining variation is explained by other factors not included in the model. Having a 68.88 percent $R^2$, our conceptualized model can be considered a good model (Hair et al. 2013). To test the significance of each hypothesized relationship, a bootstrapping procedure with a 5000 sub-sample and no-sign changes was employed. The b-values (coefficient) for each pathway are reported in the Figure 2 and Table 5. Approximately 75 percent of our study hypotheses were supported by empirical evidence at a significance level of less than 5 percent.

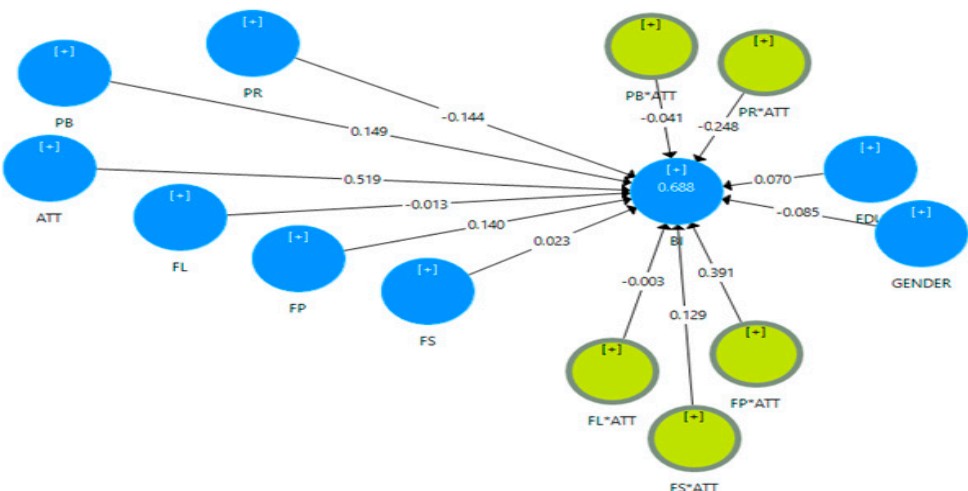

**Figure 2.** The estimated structural model with path coefficients.

**Table 5.** Path coefficients summary for direct and moderating effects.

| Hypotheses | | Coefficient | SD | *t* Statistics | Decision for Hypothesis |
|---|---|---|---|---|---|
| | | Panel A: Control Variables | | | |
| | GENDER -> BI | −0.085 | 0.067 | −1.269 | NA |
| | EDU -> BI | 0.070 | 0.022 | 3.182 ** | NA |
| | | Panel B: Exogenous Variables | | | |
| H1 | ATT -> BI | 0.519 | 0.068 | 7.632 *** | Supported |
| H2 | FL -> BI | −0.013 | 0.122 | −0.107 | Not Supported |
| H3 | FP -> BI | 0.140 | 0.066 | 2.121 ** | Supported |
| H4 | FS -> BI | 0.023 | 0.059 | 0.390 | Not Supported |
| H5 | PR -> BI | −0.144 | 0.039 | −3.692 *** | Supported |
| H6 | PB -> BI | 0.149 | 0.071 | 2.099 ** | Supported |
| | | Panel C: Moderating Terms | | | |
| H2a | FL*ATT -> BI | −0.003 | 0.059 | −0.051 | Not Supported |
| H3a | FP*ATT -> BI | 0.391 | 0.049 | 7.947 *** | Supported |
| H4a | FS*ATT -> BI | 0.129 | 0.053 | 2.434 ** | Supported |
| H5a | PR*ATT -> BI | −0.248 | 0.072 | −3.444 *** | Supported |
| H6a | PB*ATT -> BI | −0.041 | 0.106 | −0.387 | Not Supported |

EDU = education, ATT = attitude, PB = perceived benefit, PR = perceived risk, FL = financial literacy, FP = financial planning, FS = financial satisfaction, and BI = behavioral intention. 5000 sub-samples were considered in the bootstrapping procedure. ** Significant at a *p*-value < 0.05, and *** significant at a *p*-value < 0.01. SD = standard deviation; since we did not test for the demographic factor, the decisions for hypotheses were labeled as not applicable (NA).

## 4. Empirical Results Discussion

The empirical results discussed here represents the relationship nexus of investors' attitudes and behavioral intention of stock market participation in the DSE, Bangladesh. In this regard, the current study attempted to examine the moderating role of financial cognitive abilities and financial considerations on the link between attitude and behavioral intentions of investors in the DSE using the theory of planned behavior (Ajzen 1991), the theory of well-being (Wilson 1967), and the risk perception attitude theory (Rimal and Real 2003). With this aim, first, we focused on the effects of the control variables on the behavioral intentions. It can be noted that the control variables (see panel A, Table 4) had an influence on the behavioral intentions of the stock market participants. The consideration of potential extraneous variables reduced the error terms and increased statistical power (Spector and Brannick 2011). We found that individuals' educational background had a positive influence on their behavioral intention (*b* = 0.070, *t*-stat = 3.182) to participate in the stock



market, which implies that a higher education increases the likelihood of investing in the market. Such findings are expected, since individuals with good educational background are expected to be knowledgeable regarding the investment mechanism in the stock market. As a result, these individuals posed positive behavioral intentions to participating in the market.

According to the findings of this study, attitude had a positive influence on investors' behavioral intention ($b$ = 0.519, $t$-stat = 7.632) to invest in the stock market. The finding implies that investors' attitude is a significant determinant of their behavioral intentions towards stock market participation. The findings are consistent with Ajzen's (1991) theory of planned behavior, which asserts that attitude is one of the critical factors that influences specific behavior. Therefore, our findings strongly support hypothesis H1 and validate the claim of the theory of planned behavior. Previous studies have also discovered that attitude plays an important role in influencing behavioral intentions to invest in the stock market (Nadeem et al. 2020; Phan and Zhou 2014).

The study findings indicate that there were insignificant direct ($b$ = −0.013, $t$-stat = −0.107) and moderating ($b$ = −0.003, $t$-stat = −0.051) effects of financial literacy on the behavioral intentions of investors towards stock market participation. The findings suggest that financial literacy neither positively affects investor's behavioral intention towards stock market participation nor does it moderate the relationship between the attitude and behavioral intention nexus. Although the financial knowledge of the participants can influence their behavior and intentions to invest in the stock market (Hadi 2017; Kumari 2020; Al-Tamimi and Kalli 2009) and moderate the attitude and behavioral intention link (Nadeem et al. 2020), our findings are not in line with past empirical studies. Henceforth, the empirical findings do not support the study hypotheses H2 and H2a. One reason behind such findings is that the control variable, education, captured some degree of influence of financial literacy. Popat and Pandya (2018) also reported similar findings in their study on the relationship between financial knowledge and investment decisions, where the investment decisions of the investors were found to have no influence from the level of financial knowledge and some of the participants with post-graduate degrees were found to have lower levels of financial knowledge. Khalily (2016) reported that in Bangladesh, the level of financial literacy is moderate; however, to date, at the national level no survey has been conducted to determine the level of financial literacy of the country. Another plausible explanation for this type of empirical results can be due to the fact that investors utilize additional resources while taking risky financial decisions if they consider the market to be too volatile. They tend to rely on their hunches, previous experiences, and on others' recommendations instead of depending on their own financial knowledge, while implementing financial judgments in complex situations (Chen et al. 2007).

The empirical results of the study suggest that financial planning has a significantly positive effect on investors' behavioral intentions ($b$ = 0.140, $t$-stat = 2.121), which infers that better financial planning positively influences investors' decision to participate in the stock market. Arpana and Swapna (2020) and Ali (2010) stated that financial planning leads to positive financial behavior. Hence, better financial planning motivates investors to plan for the future and to make long-term investments in financial markets. In line with the empirical results, the findings of the current study provide strong support for hypothesis *H3*. Furthermore, our empirical results show that financial planning has a positive and statistically significant moderating effect on the nexus of attitude and behavioral intentions ($b$ = 0.391, $t$-stat = 7.947) towards stock market participation; therefore, the hypothesis *H3a* of this research is supported. The findings suggest that when the positive attitude and the financial planning of an investor interact, the financial planning improves the link between attitude and behavioral intention for investment decisions in the stock market. Financial planning here can be compared to the propensity to plan of Lee et al. (2019), where the study's findings suggested that the propensity to plan plays a moderating role in the financial behavior context. Investor propensity to plan is an investor's "tendency to plan for long-term goals that may result in rational, goal-setting behavior" (Ameriks et al. 2003).

Therefore, the moderating role of financial planning is supported by our hypothesis *H3a* and also by the prior empirical findings.

The statistical results of this research demonstrate that financial satisfaction has a positive but insignificant influence on behavioral intention ($b = 0.023$, *t*-stat = 0.390) towards stock market participation. Although empirical studies have revealed that financial well-being/satisfaction is an important influencer of behavioral intention towards financial decision making (Atlas et al. 2019; Joo and Grable 2004; Parmitasari et al. 2018; Yang et al. 2021), the findings of the current study imply that financial satisfaction is not a direct predictor of the variable behavioral intention towards investment decisions in the stock market. Therefore, the hypothesis *H4* of this study is not supported. The reason behind such findings could be due to the lack of confidence of investors in the stock market of the country, as reported by The Daily Star, one of Bangladesh's prominent daily newspapers (Kamal 2021). Nevertheless, the study's findings confirm that financial satisfaction positively moderates the relationship between investors' attitude and behavioral intentions ($b = 0.129$ *t*-stat = 2.434) to invest in the stock market. The results imply that when an investor's financial well-being interacts with the nexus between attitude and behavioral intention, the variable intensifies the individuals' intention to make investment decisions. Hence, the hypothesis *H4a* of this study is accepted. To achieve a desired level of financial satisfaction, individuals need to feel ready (Xiao et al. 2004) to overcome financial obstacles by healthy and desirable financial choices (e.g., saving and investing) against problematic practices (e.g., using credit cards and overspending). The positive feeling of achieving financial well-being is more likely to occur when the attitudinal factors of the individuals, for example, determination, positivity, and discipline, interact with their desired level of financial satisfaction (Ali et al. 2015). Therefore, the interaction effect of financial satisfaction with the link between investors' attitude and behavioral intentions towards stock market participation is fairly logical.

In the literature, the risk perception of investors has always been considered a significant factor in investment decision making. The empirical results of this current research suggest that perceived risk has a negative influence on the behavioral intention ($b = -0.144$, *t*-stat = $-3.692$) of investors towards their decision to participate in the market. This implies that perceived financial risk is expected to decrease investors' likelihood of making a risky investment decision in stocks. Henceforth, this finding provides statistically significant support for hypothesis *H5*. Additionally, the "risk perception attitude" theory developed by Rimal and Real (2003) advocated that risk perception is an important factor that could change behavior. Furthermore, we argue, in line with Hoque et al. (2019) and Kaur and Arora (2021), that if investors perceive the stock market to be very risky, actual returns may be lower than the estimated returns from investments, and capital loss is possible. This may cause individuals to perceive a higher level of risk. Evidently, investors might revisit their plans to invest in the stock market, which may lead to a decline in their behavioral intention to participate in the stock market. Moreover, the moderating effect of perceived financial risk on the relationship between attitude and behavioral intention of the investors towards stock market participation is negative and statistically significant ($b = -0.248$, *t*-stat = $-3.444$), which suggests that when risk perception and investors' attitude interact, it negatively affects the nexus of attitude and behavioral intentions of the individuals to make investment decisions in the stock market. Therefore, the findings of the current study and previous empirical findings (Hoque et al. 2019) support hypothesis *H5a*.

The empirical results of this research reveal that the perceived financial benefit has a positive effect on investors' behavioral intention ($b = 0.149$, *t*-stat = 2.099) to invest in the stock market. Therefore, Hypothesis *H6* is supported, which infers that the perceptions of individuals' regarding financial benefit positively influences their behavioral intentions. The RPA theory also suggests that individuals' views of benefits in relation to their decisions have a substantial impact on their likelihood of taking action (Rimal and Real 2003). Past empirical studies also revealed that one's behavioral intentions towards a financial decision are favorably influenced when they feel they will gain anything (Hoque et al. 2019).

Furthermore, past studies have highlighted the moderating role of perceived financial benefits on the link between attitude and behavioral intention for financial decision making. However, the finding of this study shows that the moderating effects of perceived financial benefits on the nexus between investors' attitude and behavioral intentions ($b = -0.041$, $t$-stat $= -0.038$) for stock market participation is not statistically significant. Hence, the hypothesis *H6a* of this research is not supported.

## 5. Summary and Implications

The main objective of this study was to examine the direct and moderating effects of financial cognitive abilities and financial considerations on the relationship between investors' attitude and behavioral intentions to participate in the stock market. The empirical findings of this study showed that investors' attitude, financial planning, and perceptions related to risk and benefits were the significant determinants of their behavioral intentions to invest in the stock market. We also identified significant moderating effects of financial planning, financial satisfaction, and financial risk perception on the relationship between attitude and behavioral intentions of the investors to participate in the stock market. The study's findings confirmed that when financial planning and financial satisfaction interact with attitude, these variables positively affect investors' behavioral intentions to invest in the market. In contrast, risk perception reduces the strength of the relationship between investors' attitude and their behavioral intentions of stock market participation.

The above presented empirical results provide a number of policy and practical implications of the current study. First, although the Bangladesh Securities Exchange Commission has launched a financial literacy program, it also should emphasize gaining the trust of investors, so that they do not feel that stock market is too volatile for investment. According to an article published in The Financial Express (Barman 2020), for the two consecutive fiscal years of 2018–2019 and 2019–2020, the net foreign investment in the DSE remained negative due to the concerns of international investors' over the country's financial sector, currency depreciation, and depressed market outlook. Often, local investors in developing countries with low levels of financial literacy follow the decisions of institutional and foreign investors in taking investment and divestment decisions in the equity market (Chen et al. 2007; Georgarakos and Inderst 2011). Therefore, the policymakers and market regulators should take proper initiatives to win back international investors' credence on the country's capital market, since the growth in the foreign investment portfolio not only plays a pivotal role in the growth of market capitalization of the country but also helps to gain the trust of local individual investors, who consider it as an important indicator of the future market outlook. Second, financial planning is a significant factor of behavioral intentions towards stock market participation, and it also intensifies the relationship between attitude and behavioral intentions. In order to encourage financial planning among individuals, the government must take initiatives to invigorate the national savings attitude (Brounen et al. 2016), which will not only safeguard them against financial shocks (Fox and Bartholomae 2020) but will also contribute towards their positive behavioral intention to participate in the stock market. Third, since investors put more importance on potential risk than benefits in taking investment decisions in the stock market, the Securities and Exchange Commission should incorporate educational modules on effective management of risk related to equity market investment in their financial literacy program. The financial service providers must also come forward in this regard and should provide financial advice to potential investors at an affordable cost to assist active and potential investors in making rational investment decisions. This might help investors avoid unnecessary risks and will make them feel positive about participating in the capital market. These initiatives are expected to shape better perceptions regarding benefits that can be gained from investment in the stock market, which will ultimately lead towards higher market capitalization. Finally, the insignificant direct effect of financial satisfaction on behavioral intention indicates that investors in the DSE do not consider the investment decision in the market rewarding enough to take the risk associated with equity investments. This

finding stipulates that market regulators and policymakers of the DSE must emphasize building the confidence of the investors, so that the investment decisions in the DSE can provide the desired level of financial satisfaction as expected by investors. Such a step will be beneficial for increasing stock market capitalization, which will ultimately positively affect the country's GDP growth rate.

This research, like many other empirical studies, has limitations. This study was carried out in a cross-sectional manner, which hinders comprehension of the intertemporal shifting pattern of investors' behavioral intentions. As a result, future studies may place emphasis on longitudinal analysis, including pre- and post-training and workshops. Furthermore, the current study was primarily focused on the stock market in Bangladesh; therefore, additional research in frontier markets, for example, Egypt, Pakistan, Nigeria, Serbia, and Sri Lanka, is required to generalize the findings. Furthermore, risk-taking tolerance behavior and technological literacy may be some interesting factors to investigate in future studies, as investment in the stock market is related to actual risk tolerance and on-line trading.

**Author Contributions:** Both the authors contributed equally towards the conceptualization, methodology, formal analysis, investigation, data curation, original draft preparation, and the writing of the manuscript. All authors have read and agreed to the published version of the manuscript.

**Funding:** This research received no external funding.

**Institutional Review Board Statement:** Not applicable.

**Informed Consent Statement:** Not applicable.

**Data Availability Statement:** Primary data for this research were collected directly from the respondents by the authors. The data will be provided upon request.

**Conflicts of Interest:** The authors declare no conflict of interest.

## Appendix A

The following questions were used for the financial literacy (FL) survey for this research. Correct answers for each question had a score of one (1) point, and for wrong answers the score was zero (0). This way, literacy levels ranged between zero (0) and ten (10).

1. If you spent BDT 70 on lunch on day 1 but only BDT 50 the next day, how much did you spend on lunch over the two days? BDT _________

2. If a prize draw win of BDT 180,000 is shared equally between six people, how much will each person receive?
BDT _________

3. If a person takes home BDT 14,000 a month and 50% of this goes to rent, what is their monthly rent?
☐     A. BDT 7000
☐     B. BDT 5000
☐     C. BDT 10,000

4. If a refrigerator priced at BDT 21,000 is discounted by 10% at a sale, how much would it cost?
☐     BDT 17,000
☐     BDT 18,900
☐     BDT 18,700

5. A deposit of BDT 2000 in a savings account earning an interest of 10% annually will become BDT_________ after 2 years.
☐     BDT 2200
☐     BDT 2420
☐     BDT 2400

6. Suppose that the interest rate on your savings account is less than the inflation rate. Using the money in the account after 1 year, you would be able to buy less goods compared to the amount you could get today.
☐     TRUE
☐     FALSE

7. If a person pays the minimum sum of his outstanding credit card balance, he will not have to pay any interest charge
☐     TRUE
☐     FALSE

8. Commonly, the outstanding balance on a credit card is subject to a financial charge (interest charge) of _____ %
☐     1% per month or 12% per annum
☐     1.85% per month or 18% per annum
☐     2% per month or 24% per annum

9. In hire purchase financing, the owner of the car is _______
☐     The hirer
☐     The banking institution
☐     Jointly between the banking institution and the hirer

10. The role of this institution is to provide advice on money management and assistance to deal with debt
A. Stock Exchange Commission
B. Bangladesh Bank
C. Credit Information Bureau

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
