# Peer review of "Moderating Effects of Financial Cognitive Abilities and Considerations on the Attitude–Intentions Nexus of Stock Market Participation"

_ijfs, doi:10.3390/ijfs10010005_

Round 1
Reviewer 1 Report
English needs to be improved. Some sentences look truncated and without a clear meaning.
It is hard to understand the original contribution of the paper. The questionnaire is built on the base of the literature and, limited to my understanding, the authors did not propose any new question and/or any new part of the questionnaire.
From a methodological point of view the authors use standard techniques.
The original part of the paper is limited to the experiment.
Author Response
Dear reviewer,
Thank you for your valuable comments. We have addressed your suggestions and have made necessary changes in the manuscript as presented in the table of the attached file.

Reviewer 2 Report
Originality:
- The motivation behind the purposed mansucript is not clear. the author(s) should clearly stated why is important to invistgate such issue. Does it have relevant policy implications? Does it help policymakers to better allocate their decisions? The same question reported in the paper, but it is no clear, and I was not able to find an answer hereinafter.
- I think the whole introduction need to be rewritten so it become informative. I see no connection in the first and the second paragraph. and I doubt how para.2 connectted with the rest of the introduction.
Relationship to Literature
what is actually the valued added of the paper with respect to the other examples that have used the same methodology? Which peculiar feature should convince the reader to believe this paper with respect to the other, inconclusive, tests? The set of countries? The methodology (which is not new) Some innovation in the econometric procedure (which is not the case)? The conclusions and, therefore, the policy implications?
For the literature reviewed in this paper, updated articles need to be included.
Overall:
The study has potential to contribute to the existing literature; however, publishing in a high-quality journal needs more clarified works. Therefore, the prevailing manuscript has needed some more revision.
Author Response
Dear Reviewer,
Thank you for your valuable comments. We have addressed your suggestions and have made necessary changes in the manuscript as presented in the table of the attached file.

Reviewer 3 Report
Dear Authors
Your research scope is interesting and the topic warrants additional evidence. Nevertheless, there are significant flaws that should be addressed to make sure your conclusions are sound:
1 - Please consider a professional editing of your paper. As is, it lacks flow and shows several typo;
2 - Reconsider how the different theories can be applied to your research question. In your current version, you are choosing some of the features of TPA, leaving others out (subjective norm, for instance). Then choosing other theory, as if you were doing "cherry Picking", rather than framing consistently your hypotheses.
3 - Please provide a preliminary analysis of the internal validity of your survey, including further details of the process of collection of the answers, potential common method biases, and overall reliability.
4 - You do not present any information about the items used for each construct (except for financial literacy). Please provide them transparently.
5- I also miss information on why and how you mix items from different previous research papers for the same construct. This is potentially harmfull to construct validity.
6 - Furthermore, your financial literacy items are not the most commonly used in the literature (Lusardi and others). Why? These questions are measuring financial literacy mostly at a very low level.
7- Your measurement model lacks significant details. How did you construct it? Reflexive or formative? What are the extrated factors for each construct?
This concerns, if addressed, i fell they will increase the quality, consistency and impact of your research.
Kind regards
Author Response

(The authors gave the same response as above.)

Round 2
Reviewer 1 Report
The paper has been substantially revised highlighting the main contribution.
Author Response
Dear Reviewer,
Thank you for your valuable comments.
Reviewer 3 Report
Dear Authors
Congratulations on the significant improvement of your paper. Minor typo and formating issues subsist.
Interesting paper.
Kind regards
Author Response

(The authors gave the same response as above.)
